# Preparation and Application of a New Two-Component Superhydrophobic Coating on Aluminum Alloy

Chao Qiu *, Shuai Liang, Meng Li, Han Cheng and Wenfeng Qin

Aviation Engineering Institute, Civil Aviation Flight University of China, Guanghan 618300, China; liangshuai19990801@163.com (S.L.); mokechi@163.com (M.L.); chenghanstorm@sina.com (H.C.); qwfgrh@126.com (W.Q.)
* Correspondence: chaoqiu1987@163.com

**Abstract:** Superhydrophobic surfaces have been widely used for their corrosion resistance, self-cleaning and anti-icing characteristics. A new two-component superhydrophobic coating was prepared on aluminum alloy, and some application properties were studied. With appropriate silica, the contact angle of the two-component superhydrophobic coating can be 164.4°, and it has good resistance to the continuous hitting of water droplets and the corrosion of acid. Even when it had been continuous impacted by acid droplets for 300 min, the contact angle of the coating was still lager than 150°. However, the coating was easily corroded by sodium hydroxide. Moreover, it can not only reduce its freezing point by more than 5 °C, but also delay the freezing of droplets on aluminum alloy by about 20 s at the temperature of −20 °C. More than that, the growth of ice or frost on it can only cause extremely minor mechanical damage to it.

**Keywords:** two-components; superhydrophobic coating; droplets hitting; corrosion; anti-icing

## 1. Introduction

Superhydrophobic surfaces have been widely used for their corrosion resistance, self-cleaning and anti-icing characteristics, especially in some cold areas. The superhydrophobic hull of ship can not only reduce the resistance of water, but also prevent the adhesion of halobios such as mussels or barnacles [1,2]. Superhydrophobic pipeline can prevent petroleum from adhering to the wall of pipeline and reduce its consumption during transportation [3]. In addition, a superhydrophobic transmission line can realize anti-icing by delaying droplet freezing [4].

In most research, a superhydrophobic surface was prepared by imitating some natural superhydrophobic surfaces, such as lotus leaf, locust leaf and nepenthes [5,6]. For example, Liu prepared a material by imitating the locust leaf. The resultant material surface has the characteristics of superhydrophobicity and low adhesion, the contact angle is more than 150°, the sliding angle is about 6.6°, and it possesses excellent self-cleaning performance [7]. Inspired by poison dart frog, Sun prepared an efficient superhydrophobic coating consisting of an outer porous superhydrophobic epidermis and a wick-like underlying dermis that was infused with anti-freeze liquid. The coating delays the onset of glaze formation ten times longer than surfaces flooded with a thin film of antifreeze [8].

Surface modification is an effective way to make metal or nonmetal material superhydrophobic [9,10]. However, different materials have different technologies for modification, and the technologies of modification are very difficult, which make the cost of superhydrophobic surfaces obtained by modification extremely expensive. It seriously restricts the application and development of superhydrophobic surface [11,12]. On the other hand, low-surface-energy coatings containing only one component are usually applied in engineering [13], such as polytetrafluoroethylene, which can be easily prepared and painted on most materials by airbrushes and make it hydrophobic [14]. Nevertheless, the hydrophobicity of these coatings is much worse than that of the modified surface.

In that case, a new two-component superhydrophobic coating containing fluorine and nano particles was prepared by a simple process. The coating could be easily painted on most materials and also make it superhydrophobic. Then, some application properties of the coating were studied as well.

## 2. Preparation of the New Two-Component Superhydrophobic Coating

Three grams of silica was added to 40 mL of anhydrous ethanol, and a drop of KH550 organic solvent was added as well, which could improve the dispersion of silica in the polymer. The diameter of the silica was 100 nm. In order to disperse silica in solution, a magnetic stirrer was used to stir the solution for 15 min. Then the solution was put into a vacuum drying oven and kept at 60 °C for 24 h until the anhydrous ethanol evaporated completely and the modified silica left only.

Amounts of 0.2 g, 0.5 g, 0.8 g and 1.1 g modified silica were added to 4 cups of 10 mL deionized water, respectively, and all solutions were stirred for 15 min by the magnetic stirrer. Five milliliters of PTFE was added to every solution afterward and stirred for another 15 min. The temperature and speed of magnetic stirrer were 30 °C and 300 RPM, respectively. At this point, the preparation of the two-component coating had been completed. The whole process is shown in Figure 1.

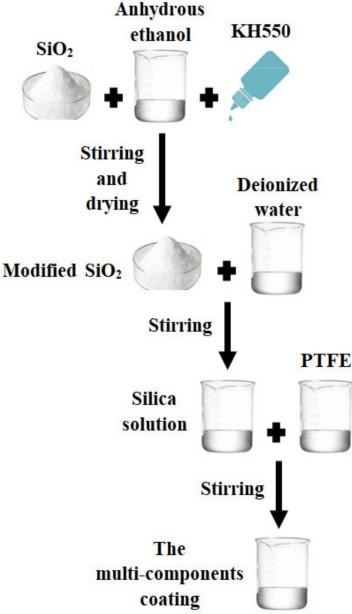

**Figure 1.** Preparation of the two-component coating.

In this work, 7075 aluminum alloy was used as substrate, the size of which was 30 mm × 5 mm × 1.5 mm. The main alloying element of 7075 aluminum alloy was zinc, it was widely used in aviation. The two-component coatings were sprayed onto the surfaces of several identical 7075 aluminum alloys by spraying devices as shown in Figure 2 and were dried at ambient temperature of 26 °C for 24 h. Thickness of the coating was 85 μm. The contact angles and sliding angles of all coatings have been measured by contact angle measurement, it shown in Table 1.

In Table 1, the contact angle of the coating containing 0.2 g silica is less than 150°. That means the coating is not superhydrophobic. Except for this, all of the other coatings are superhydrophobic. In these coatings, the coating with 0.8 g silica has the largest contact angle and the least sliding angle, therefore, the hydrophobicity of which is the best.

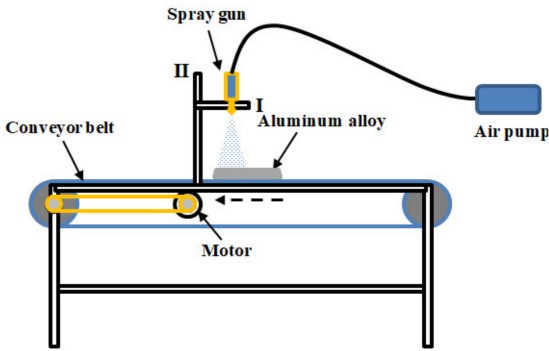

**Figure 2.** Spraying devices.

**Table 1.** Contact angles and sliding angles of all the coatings.

| Components of Coating | Contact Angle | Sliding Angle |
|---|---|---|
| 0.2 g $SiO_2$ + 10 mL $H_2O$+ 5 mL PTFE | 145.2° | — |
| 0.5 g $SiO_2$ + 10 mL $H_2O$+ 5 mL PTFE | 156.5° | 5.4° |
| 0.8 g $SiO_2$ + 10 mL $H_2O$+ 5 mL PTFE | 164.4° | 1° |
| 1.1 g $SiO_2$ + 10 mL $H_2O$+ 5 mL PTFE | 154.4° | 7° |
| Without coating | 74.3° | — |

## 3. Continuous Hitting of Water Droplets on the Two-Component Superhydrophobic Coating

Devices of the experiment are shown in Figure 3. The water flow of the microinjection pump was 100 mL/h. In that case, water droplets formed at the tip of syringe one by one, and then fell and hit the aluminum alloy coated with two-component superhydrophobic coating. The falling height of droplets was 40 mm, and the diameter of droplets was 2.2 mm. The hitting of droplets lasted 30 min. Every droplet hit and rebounded off the coated aluminum alloy. It is shown in Figure 4. The frame rate of the high-speed camera was 2/3 ms.

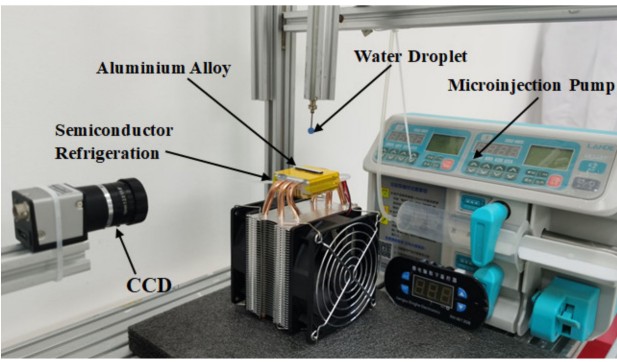

**Figure 3.** Devices of the hitting experiment.

After hitting the two-component superhydrophobic coating, with the help of inertial force, the droplet could overcome the surface tension, dynamic friction and adhesion between droplet and coating, which made the droplet spread radially in the form of surface wave. When the droplet extended to the maximum spreading surface, the surface tension of the droplet was much greater than that in the static state. The droplet was unstable. Then with the help of surface tension, the droplet overcame the dynamic friction and adhesion and began to retract. The flattened droplet was squeezed by surface tension and the surrounding liquid, the center of which rose gradually and stretched into a pushpin. With the help of surface tension and inertial force, the droplet kept rising, and then changed into a bowling bottle with slender bottom and round top [15]. The top of the droplet

continued to move upward, while the bottom of which did not leave the coating due to the adhesion. In that case, two secondary droplets appeared for the break of droplet. The smaller secondary droplet continued to move upward, and the shape of which varied between spherical and ellipsoidal due to the instantaneous impulse when droplet break. Moreover, with the help of instantaneous impulse and inertial force, the larger secondary droplet left the coating finally and moved upward in a peristaltic way [16].

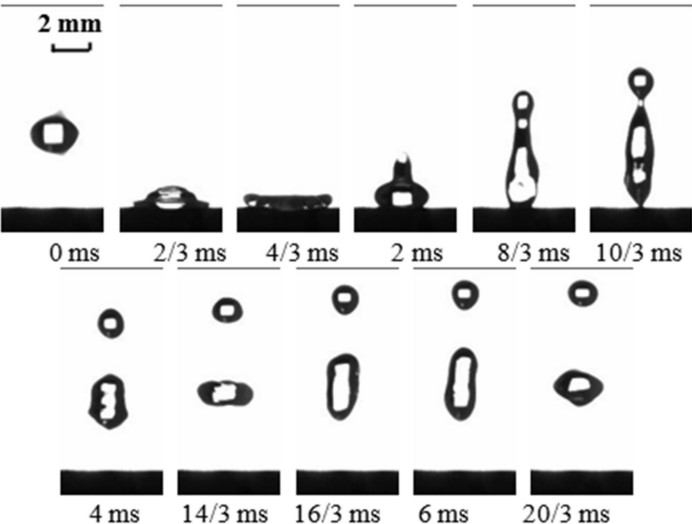

**Figure 4.** Hitting and rebounding of a droplet on the coated aluminum alloy.

Most research indicates that the rebound height of droplets was closely related to the hydrophobicity of the coating [17]. In general, the poorer the wettability of the coating surface, the higher the rebound height of the droplets. Therefore, it was necessary to study the rebound heights of droplets at different moments of the hitting experiment, which could indicate the effect of continuous hitting on the two-component superhydrophobic coating. It is shown in Figure 5.

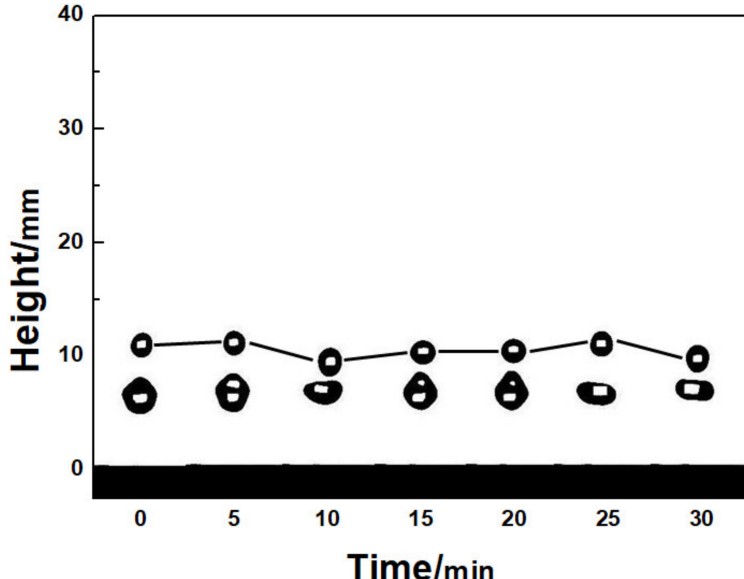

**Figure 5.** Rebound heights of the droplets at different moments.

Although the two-component superhydrophobic coating was continuously impacted by water droplets, there was little change in the rebound height of the larger secondary

droplets. The rebound height of the smaller secondary droplets fluctuated between 9.67 mm and 11.07 mm, and there was no significant increase or decrease. On the other hand, after being impacted by continuous water droplets for 30 min, the contact angle on the coating only decreased from 163.4° to 163.0°. According to the analysis of rebound height and contact angle, the continuous hitting of water droplets did not cause any damage to the two-component superhydrophobic coating. This is due to the silica in the coating. The silica is hard enough to resist the continuous hitting of water droplets. Moreover, air trapped in a rough structure of the coating can absorb the shock of hitting droplets partly [18].

## 4. Corrosion of the Two-Component Superhydrophobic Coating by Acid and Alkali

Devices of the corrosion experiment are shown in Figure 6. The liquid flow of the microinjection pump was 100 mL/h. In that case, acid or alkali droplets formed at the tip of syringe one by one, and then rolling along the aluminum alloy coated with two-component superhydrophobic coating. Sulfuric acid, hydrochloric acid and nitric acid were used as the experiment solution, respectively, which were the essential component of acid rain [19]. The sodium hydroxide, a corrosiveness alkali, was used as the experiment solution as well. The concentrations of all solutions were 1 mol/L. The falling height of droplets was only 5 mm, avoiding droplets bouncing on the coating. The diameter of droplets was 2.2 mm. End of the aluminum alloy was laid on a platform with the thickness of 2 mm. In that case, the aluminum alloy was sloped, and the angle between aluminum alloy and semiconductor refrigeration was only 4°. The experiment had last 300 min. It is shown in Figure 7.

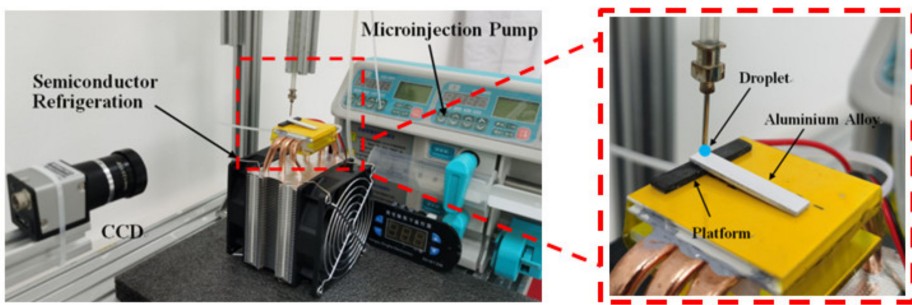

**Figure 6.** Devices of the corrosion experiment.

In Figure 7, all of the sulfuric acid droplets, hydrochloric acid droplets and nitric acid droplets could still roll along the coatings by their own gravity though the continuous droplets had rolled on each coating more than 300 min. However, there were some troubles for the sodium hydroxide droplets. When the experiment had last 100 min, a sodium hydroxide droplet stopped on the coating suddenly. It stayed on the coating for a moment until the next droplet rolled towards to it and pushed it away. Unfortunately, the second droplet was trapped on the same place as well. Such an accident was happened again and again, which indicated that the coating had been destroyed here. The contact angles on each coating after the corrosion experiment were list in Table 2.

**Table 2.** The contact angles on each coating before and after the acid or alkali corrosion.

| Acid or Alkali | Concentration | Time of Corrosion | Contact Angles | |
| --- | --- | --- | --- | --- |
| | | | Before Corrosion | After Corrosion |
| Sulfuric acid | 1 mol/L | 300 min | 163.4° | 159.6° |
| Hydrochloric acid | 1 mol/L | 300 min | 163.2° | 159.0° |
| Nitric acid | 1 mol/L | 300 min | 165.5° | 159.5° |
| Sodium hydroxide | 1 mol/L | 100 min | 164.3° | 143.7° |

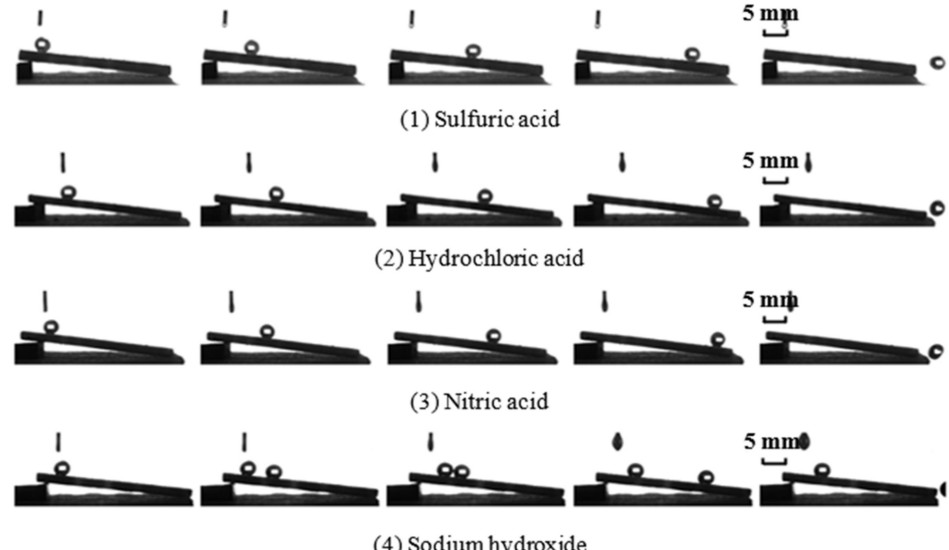

**Figure 7.** Rolling of acid or alkali droplets on the aluminum alloys coated with two-component superhydrophobic coating.

It could be seen that after the corrosion of sulfuric acid, hydrochloric acid or nitric acid, all of the contact angles on the coatings decreased slightly. It is well known that neither PTFE nor silica in the coating will react with sulfuric acid, hydrochloric acid or nitric acid. The slight decrease in the contact angle may be cause by a very small amount of liquid trapped in the rough structure when droplets rolling along the coating. Nevertheless, the contact angle of the coating was still more than 150°, thus the coating was superhydrophobic as well. Therefore, the two-component superhydrophobic coating has a good resistance to the corrosion of sulfuric acid, hydrochloric acid and nitric acid, which can prevent the materials from being corroded by acid rain.

On the other hand, the contact angle on the coating corroded by sodium hydroxide decreased significantly. It was less than 150°, thus the coating was no longer superhydrophobic. That is because sodium hydroxide can react with silica, and sodium silicate is formed. In that case, the silica in the coating was gradually consumed, and the rough structure of the coating was destroyed at the same time, although the contact surface between sodium hydroxide droplets and the coating was extremely small. SEM images of the two-component coating before and after the corrosion of sodium hydroxide were shown in Figure 8.

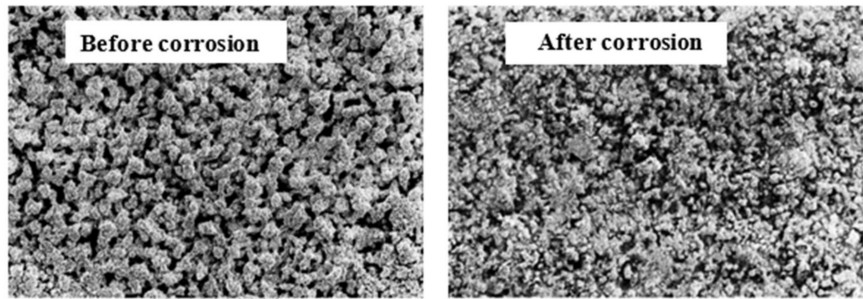

**Figure 8.** SEM images of the two-component coating before and after the corrosion of sodium hydroxide.

Obviously, the rough structure built by silica had disappeared partly on the coating. Instead, some corrosion morphology was observed on the coating. Therefore, the two-component superhydrophobic coating has a poor resistance to the sodium hydroxide.

## 5. Icing and Frosting on the Two-Component Superhydrophobic Coating

Devices of the freezing experiment are shown in Figure 9. An aluminum alloy coated with the two-component superhydrophobic coating was placed on the semiconductor refrigeration. Another aluminum alloy without coating was also placed on the semiconductor refrigeration the same time. The temperature of the semiconductor refrigeration was controlled by a temperature sensor. The growth of ice and frost on the two aluminum alloys was recorded by a high-speed camera.

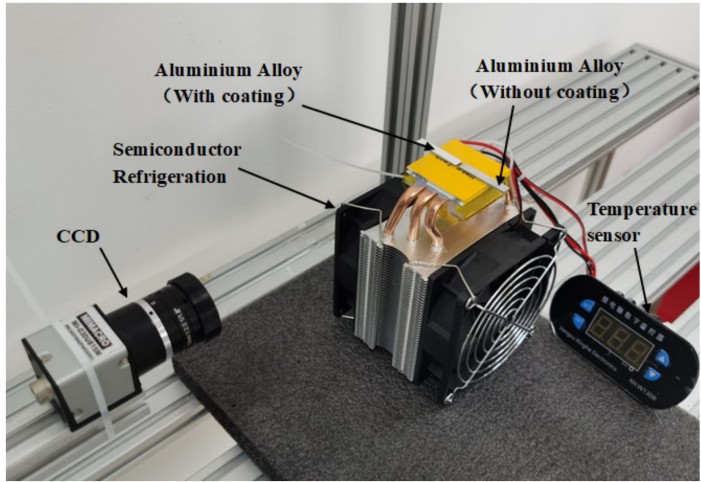

**Figure 9.** Devices of the freezing experiment.

### 5.1. The Growth of Frost on the Two Aluminum Alloys

The ambient humidity was 50%, the temperature of the semiconductor refrigeration was set to −5 °C. The growth of frost on the two aluminum alloys was shown in Figure 10.

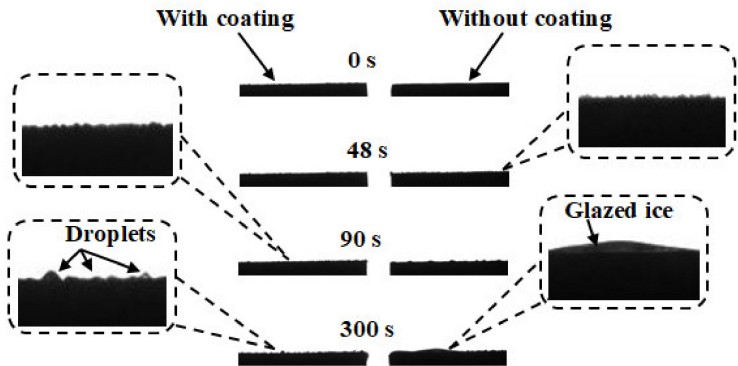

**Figure 10.** The growth of frost with the temperature of −5 °C.

It could be seen that some small droplets were condensed on the uncoated aluminum alloy about 48 s later. Then, the droplets gradually converged and froze. Finally, glazed ice was covered on the uncoated aluminum alloy. On the coated aluminum alloy, there were no small droplets until 90 s, which was later than that on the uncoated aluminum alloy. In addition, the droplets on the coated aluminum alloy kept liquid instead of freezing. Therefore, the two-component superhydrophobic coating can not only delay the condensation of air in ambient on its surface, but also reduce the freezing point more than 5 °C. On the other hand, it was found that there would be no frost on both aluminum alloys with the temperature of −5 °C. That is because the thermodynamic barrier of frosting is larger than icing, the temperature of −5 °C is not cold enough to overcome the thermodynamic barrier of frosting, though the temperature can make the nucleation of ice nucleus in droplets. In that case, the air in ambient conditions can only be condensed

into droplets and be frozen later on uncoated material or be kept liquid at all times on the coated material. When the temperature of the semiconductor refrigeration was set to −20 °C, the growth of frost on the two aluminum alloys was as shown in Figure 11.

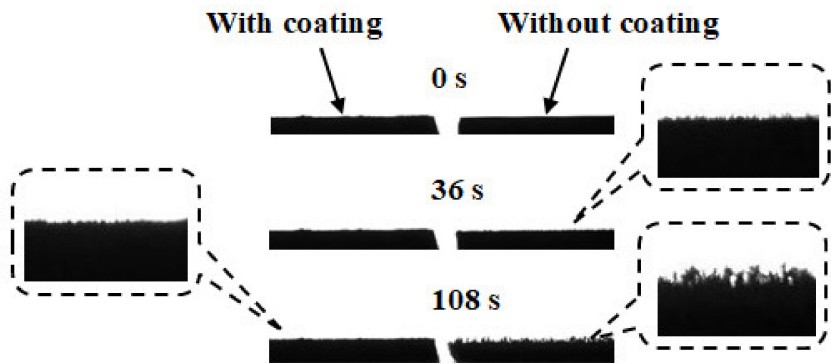

**Figure 11.** The growth of frost with the temperature of −20 °C.

In Figure 11, frost appeared on the uncoated aluminum alloy about 36 s later, while this did not happen on coated aluminum alloy until 108 s. At that moment, the frost on the uncoated aluminum alloy had grown splendidly. Therefore, the two-component superhydrophobic coating can delay the frosting on aluminum alloy about 72 s at the temperature of −20 °C.

*5.2. Freezing of a Droplet on the Two Aluminum Alloys*

Two identical droplets were placed on the two aluminum alloys, respectively. The temperature of the semiconductor refrigeration was set to −5 °C, and freezing of droplets was as shown in Figure 12.

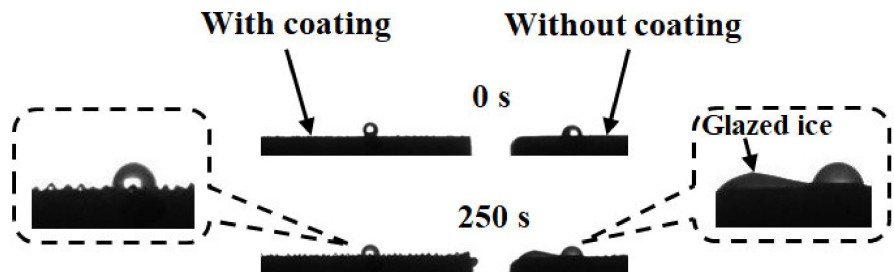

**Figure 12.** Freezing of a droplet on aluminum alloys with the temperature of −5 °C.

In the figure, the droplet on the uncoated aluminum alloy froze about 250 s later, and the frozen droplet was hemispherical. Not only that, glazed ice also appeared on the uncoated aluminum alloy, which was the result of the condensation of air in ambient on the aluminum alloy. On the other aluminum alloy, the droplet was still liquid but the contact angle on aluminum alloy was significant reduced. This is due to the condensation of air trapped in rough structure of the two-component superhydrophobic coating at low temperature. In addition, there were lots of smaller droplets appearing on the coated aluminum alloy, which were liquid as well. Therefore, it is indicated again that the freezing point of droplets on the aluminum alloy can be reduced at least 5 °C by the two-component superhydrophobic coating. When the temperature of the semiconductor refrigeration was set to −20 °C, the freezing of droplets was as shown in Figure 13.

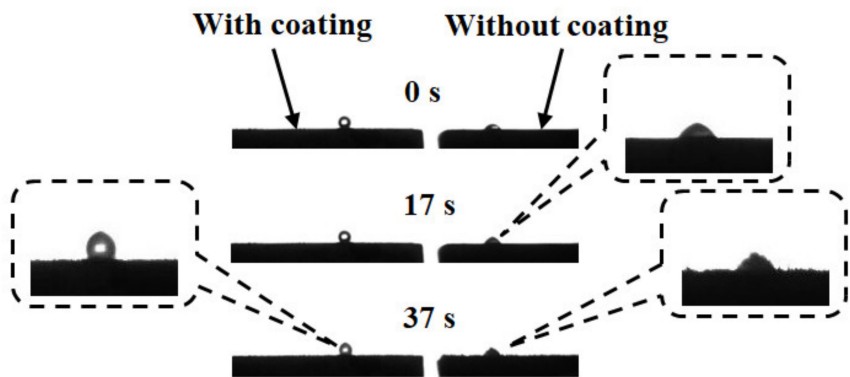

**Figure 13.** Freezing of droplets on aluminum alloys with the temperature of −20 °C.

It could be seen the droplets on the uncoated aluminum alloy froze about 17 s later, and the frozen droplet was shaped like a mountain. Then, some frost grew on it. On the other hand, the droplet on the coated aluminum alloy froze about 37 s later, which looked like a peach. The formation of this singular shape originated from the reduction in mass density during the freezing process [20]. Therefore, the two-component superhydrophobic coating can delay the freezing of droplets on aluminum alloy about 20 s at the temperature of –20 °C. It is because the superhydrophobic coating can enlarge the thermodynamic barrier of freezing, preventing the nucleation of ice nucleus in droplet, thus reducing the freezing point of the droplet, and delaying the freezing of the droplet [21]. Moreover, thermal insulation of the air trapped in the rough structure of the superhydrophobic coating reduced the heat loss of the droplet as well.

### 5.3. The Effect of Icing and Frosting on the Two-Component Superhydrophobic Coating

In order to study the damage of icing or frosting to the two-component superhydrophobic coating, the coated aluminum alloys were dried after icing or frosting. The contact angles on coating are shown in Table 3.

**Table 3.** The contact angles on each coating before and after the freezing or frosting.

| Type | Temperature | Humidity | Duration of Freezing or Frosting | Result | Contact Angles | |
| | | | | | Before | After Drying |
|---|---|---|---|---|---|---|
| Frosting | −20 °C | 50% | 10 min | Frosted | 160.1° | 155.5° |
| Icing | −20 °C | 50% | 10 min | Frozen | 160.0° | 157.4° |

It could be found that the contact angles on the two-component superhydrophobic coating only decreased slightly after the freezing or frosting. This is due to the excellent wear resistance of silica and PTFE in the coating [22]. In addition, all of the silica in the coating are nano particles. Even if some of the silica is ground off, the rest can still build the rough structure of superhydrophobic surface. Therefore, freezing or frosting can only make an extremely minor mechanical damage to the two-component superhydrophobic coating.

### 6. Conclusions

A new two-component superhydrophobic coating was prepared on aluminum alloy, some application properties of which were studied as well. It is concluded as follows:

(1) The two-component superhydrophobic coating has a good resistance to water droplets hitting. Continuous falling droplets can still rebound off the two-component superhydrophobic coating after hitting it for 30 min, and will not cause any damage to the two-component superhydrophobic coating.

(2) The two-component superhydrophobic coating has a good resistance to most acids, such as sulfuric acid, hydrochloric acid and nitric acid. Therefore, it can prevent the aluminum alloy from being corroded by acid rain. However, the two-component

superhydrophobic coating has a poor resistance to the sodium hydroxide, the rough structure of which is easy to be corroded.

(3) The two-component superhydrophobic coating can not only reduce the freezing point on it more than 5 °C, but also delay the growth of ice and frost on it. On the other hand, the growth of ice or frost will make an extremely minor mechanical damage to the two-component superhydrophobic coating.

**Author Contributions:** Conceptualization, C.Q.; methodology, M.L.; formal analysis, H.C. and W.Q.; investigation, S.L.; writing—original draft preparation, C.Q. All authors have read and agreed to the published version of the manuscript.

**Funding:** The Natural Science Foundation of China (No. 11602293), the State Key Laboratory of Icing and Anti/De-icing Research fund (No. IADL 20190407), the General Program of CAFUC (No.BJ2016-06).

**Institutional Review Board Statement:** Not applicable.

**Informed Consent Statement:** Not applicable.

**Data Availability Statement:** The study did not report any data.

**Acknowledgments:** This work was supported by CAAC and Aeroengine Operation Safety and Control Technology Research Center. In addition, we are particularly grateful to Professor Shang Yongfeng for his help in our research.

**Conflicts of Interest:** The authors declare no conflict of interest.

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
