# Peer review of "Preparation and Application of a New Two-Component Superhydrophobic Coating on Aluminum Alloy"

_metals, doi:10.3390/met12050850_

Round 1
Reviewer 1 Report
In my opinion, the article is well thought out and ready for printing.
Author Response
Thank you for your guidance and advice

Reviewer 2 Report
- The coating described in the article would be more correctly called not multi-component, but two-component, since it contains 2 components (silica and PTFE).
- The annotation should briefly describe the composition of the coating and the method of its preparation.
- It would be necessary to reveal the essence of the process of modifying silica with the help of an organic solvent KH550.
- Nothing is said about the technology and equipment for spraying coating on aluminum alloy.
- The composition of aluminum alloy 7075 and the dimensions of samples from this alloy should have been given.
- You should not talk about two or several identical 7075 aluminum alloys. It is better to talk about bare (uncoated) and coated samples.
- It would be necessary to provide data on the thickness and roughness of the obtained coatings.
- Methods for measuring the contact angle and slip angle should be given or referred to.
- Figure 2 - defective.
Author Response
- It has been revised.
- It has been revised.
- It has been revised.
- It has been revised.
- It has been revised.
- It has been revised.
- It has been revised.
- It has been revised.
- It has been revised.

Reviewer 3 Report
Please address all comments in the attahced file. But there are some general comments here:
1- Please modify abstract by using more data.
2- Please revise the introduction according the aim and scopes.
3- Please provide more quality figures.
4- Please provide a comparison of different SiO2 content surfaces.
5- Please specify why classical corrosion tests was not applied.

Author Response
1.More data have been added in abstract.
2.More references have been added in introduction according the aim and scopes.
3.Coating thickness has been provided in this work.
4.Comparison of contact angles of different SiO2 content surfaces has been shown in table 1. As we know, the greater the contact angle, the better the hydrophobic and anti-icing of the coating. Therefore, we chose the coating with the largest contact angle to study the corrosion and anti-icing characteristics of the coating.
5.The corrosion test used in this paper is to simulate the corrosion of falling acid rain on the coating surface and sliding along the coating surface.Thank you for your suggestion. We will continue to explore the corrosion of classical corrosion on the coating surface.

Round 2
Reviewer 3 Report
The comments are addressed.